# Predicting scalar diversity with context-driven uncertainty over alternatives

**Jennifer Hu[1], Roger Levy[1], Sebastian Schuster[2]**
[1]Department of Brain and Cognitive Sciences, Massachusetts Institute of Technology
[2]Department of Linguistics & Center for Data Science, New York University
`jennhu@mit.edu, rplevy@mit.edu, schuster@nyu.edu`

## Abstract

Scalar implicature (SI) arises when a speaker uses an expression (e.g., *some*) that is semantically compatible with a logically stronger alternative on the same scale (e.g., *all*), leading the listener to infer that they did not intend to convey the stronger meaning. Prior work has demonstrated that SI rates are highly variable across scales, raising the question of what factors determine the SI strength for a particular scale. Here, we test the hypothesis that SI rates depend on the listener's confidence in the underlying scale, which we operationalize as uncertainty over the distribution of possible alternatives conditioned on the context. We use a T5 model fine-tuned on a text infilling task to estimate this distribution. We find that scale uncertainty predicts human SI rates, measured as entropy over the sampled alternatives and over latent classes among alternatives in sentence embedding space. Furthermore, we do not find a significant effect of the surprisal of the strong scalemate. Our results suggest that pragmatic inferences depend on listeners' context-driven uncertainty over alternatives.

## 1   Introduction

Human communication involves not only the transmission of linguistic signals, but also context-guided inference over the beliefs and goals of other conversational agents (e.g., Sperber and Wilson, 1986; Grice, 1975). One signature pattern of this pragmatic reasoning is scalar implicature (SI). The standard view is that SIs arise as a result of ordered relationships between linguistic items – when a weaker (less informative) item of a scale is uttered, then a listener can infer that the speaker did not have grounds to utter the stronger (more informative) item on that scale. For example, if Alice says "Some of the students passed the exam", Bob can draw the scalar inference that *not all* students passed the exam, even though Alice's utterance would still be semantically true in that scenario.

While this view predicts that SIs are context-independent and generally strong – known as the Homogeneity Assumption (Degen, 2015) – empirical studies have demonstrated a remarkable amount of variance in SI rates both within (Degen, 2015; Li et al., 2021) and across lexical scales (Doran et al., 2009; van Tiel et al., 2016; Gotzner et al., 2018; Pankratz and van Tiel, 2021). This raises the question of what factors determine the SI strength for a particular scale. In a landmark study, van Tiel et al. (2016) test two classes of potential predictors of SI strength: the availability of the strong scalemate given the weak scalemate, and the degree to which scalemates can be distinguished from each other. They demonstrate that availability is not a reliable predictor of SI strengths (but see Westera and Boleda 2020), while measures of scalemate distinctness, such as the boundedness of the scale, do robustly predict SI. More recent studies (e.g., Gotzner et al., 2018; Sun et al., 2018; Pankratz and van Tiel, 2021; Ronai and Xiang, 2022) have proposed a variety of other factors such as negative strengthening, polarity, and extremeness.

Here, we revisit the hypothesis that SI rates depend on the availability of the strong scalemate. While prior work has operationalized availability with measures of the strong scalemate such as word frequency or similarity/association with the weak scalemate (van Tiel et al., 2016; Westera and Boleda, 2020; Ronai and Xiang, 2022), we re-frame availability as the listener's *confidence in the underlying scale*. Upon hearing a scalar expression, listeners must determine the items on the scale as well as the ordering metric before inference proceeds (Hirschberg, 1985). If the listener is less certain about the scale, then they will be less likely to exclude the meaning of a particular strong scalemate. We operationalize scale uncertainty as uncertainty over the alternatives that could serve as a strong scalemate to the observed scalar expression. To estimate the alternatives predicted

by humans, we use a T5 model (Raffel et al., 2020) fine-tuned on a text infilling task. While prior studies have treated alternatives as linguistic forms, we also consider the idea that listeners reason about alternatives at a conceptual level (Buccola et al., 2021) by treating alternatives as latent classes in a conceptual space. Our results support the role of scale uncertainty in determining SI rates, and suggest a new way of testing conceptual theories of alternatives for scalar inference.

## 2 Human data

To obtain human SI strengths, we use the data from Experiment 2 by van Tiel et al. (2016). In our analyses, we only consider the adjectival scales from van Tiel et al.'s original materials, resulting in 32 scales. Each scale is a pair of adjectives ⟨[WEAK], [STRONG]⟩, where the meaning of [STRONG] entails the meaning of [WEAK] (e.g., ⟨*intelligent, brilliant*⟩). The experiment measures whether humans exclude the meaning of [STRONG] upon observing a speaker use [WEAK].

On each trial of the experiment, participants read a prompt of the form "John says: [NP] is [WEAK]", where [WEAK] is an adjective scalar item that may trigger a scalar inference, and [NP] is a noun phrase that sets the context for the scalar item. There were 3 such sentences per scale, which differ from each other only in the NP. For example, the weak scalar item *intelligent* is associated with the sentences "This student/That professor/The assistant is intelligent". Participants were then asked: "Would you conclude from this that, according to John, [NP]$_P$ is not [STRONG]?", where [STRONG] is the strong scalemate to [WEAK], and [NP]$_P$ is a pronominalized version of the [NP] in the speaker's original utterance (e.g., "she is not brilliant"). Participants marked their response as Yes or No. The SI rate for a scale is computed as the proportion of Yes responses averaged over participants and sentences.

## 3 Predictors

We use T5 (Raffel et al., 2020) to estimate all probabilities in our analyses. T5 is a sequence-to-sequence Transformer model (Vaswani et al., 2017) trained to represent language processing tasks as text-to-text problems. Our model is based on the pre-trained T5-base model from Hugging-face Transformers (Wolf et al., 2020). Since the off-the-shelf T5 model is not optimized for text generation, we use a T5 model that has been fine-tuned

on a text infilling task (Qian and Levy, 2022). The model is fine-tuned on a 10-million-token subset of the 2007 portion of the New York Times Corpus (Sandhaus, 2008). The supervision signal is generated by randomly masking some spans of words in a sentence to get the fragmentary context and a plausible completion. At inference time, the model decodes autoregressively via greedy sampling.

### 3.1 Predictability of strong scalemate

As a baseline, we first consider whether SI rates – i.e., the rate at which [WEAK] is taken to exclude the meaning of [STRONG] – are explained by the context-conditioned predictability of the tested strong scalemate. This is similar to production-based measures of availability, such as the tendency of humans to mention the strong scalemate in a Cloze task (van Tiel et al., 2016; Ronai and Xiang, 2022). However, these metrics are expensive to estimate, especially if we wish to estimate the full distribution of alternatives. We address this by using T5 as a proxy of human predictions, taking the view that humans maintain expectations about possible alternatives via a predictive language model optimized on the surface statistics of language.

To measure the predictability of a certain linguistic expression as a strong scalemate under T5, we leverage scalar constructions (Hearst, 1992; de Melo and Bansal, 2013; Pankratz and van Tiel, 2021). Scalar constructions are patterns such as *X, but not Y*, which indicate a scalar relationship between a weak item $X$ and strong item $Y$. For each weak scalar item in our test materials, we construct a scalar template of the following form:

$$[\text{NP}] \text{ is } [\text{WEAK}], \text{ but not } \underline{\qquad}. \qquad (1)$$

We have 3 such templates for each scale, where [NP] is given by the 3 sentences from van Tiel et al.'s materials. By embedding the weak scalar item within the *X, but not Y* construction, the model should set up expectations for a potential scalemate in the masked position. For each ⟨[WEAK], [STRONG]⟩ pair from van Tiel et al.'s items, we substitute the strong scalemate into the masked position and compute the surprisal (i.e., negative log probability) at that token under T5.[1] Language model surprisal has been shown to predict psychometric measures of human sentence processing (e.g., Smith and Levy, 2013; Goodkind and Bicknell, 2018; Wilcox et al., 2020), suggesting that

---

[1] When scalar items are split into multiple tokens, we obtain surprisals by summing over these sub-word tokens.

the distribution learned by these models captures expectations deployed by humans during real-time language comprehension.

## 3.2 Scale uncertainty

Next, we test the hypothesis that SI depends on the listener's uncertainty about the scale implied by the speaker's utterance. Depending on the context, a single word (e.g., *bad*) could lie on multiple scales – e.g., "The food is bad" might imply that the food is not rotten, whereas "The score is bad" might imply that the score is not failing. This uncertainty is not a function of a particular scalemate (unlike the availability measure described in Section 3.1 and in prior work), but rather a property of the scalar trigger and the context in which it is observed.

We operationalize scale uncertainty as uncertainty over the distribution of possible alternatives conditioned on the context. To obtain a set of candidate alternatives $A$, we sample $N = 100$ completions from the T5 infilling model given the scalar template in Equation (1).[2] During decoding, we restrict the maximum number of generated tokens to 5, and only keep the unique completions. We further process the outputs by removing punctuation and casing, and only keep the first word of the sequence (e.g., "always" and "always," would be collapsed into "always"). After this step, we also removed completions that consisted only of stop-words.[3] We performed these processing steps in order to reduce the sensitivity of the model-generated alternatives distribution to low-level features like punctuation, and to account for the model's tendency to output high-frequency function words.

## 3.3 Strings vs. concepts

For each of our surprisal and scale uncertainty measures, we consider two operationalizations that reflect differing theories of alternatives. The first assumes that surface-level linguistic forms (i.e., strings) are the alternatives driving SI. The second view is that listeners reason about alternatives at a conceptual level (Buccola et al., 2021), which we estimate using sentence embeddings.

**String-based measures.** We first consider the string-based view of alternatives. We obtain string-based surprisal by plugging the strong scalemate

into the blank in Equation (1) (i.e., $Y$ in the *X, but not Y* construction) and computing its context-conditioned surprisal under T5. Similarly, to obtain a string-based measure of scale uncertainty, we compute uncertainty over the strings that fill the masked position in the scalar template (Equation (1)). That is, we normalize the probabilities of each $a \in A$ to obtain a probability distribution over alternatives, and then compute the Shannon entropy over this distribution. We predict that lower surprisal reflects a more predictable alternative, and thus results in a stronger SI. Similarly, lower entropy reflects lower uncertainty over the underlying scale, and should lead to a stronger SI.

This method implicitly assumes that surface-level linguistic forms (i.e., strings) are the alternatives driving scalar inferences. As a single concept can be expressed with multiple forms, however, the surprisal over forms may not be a good estimate of the surprisal of the underlying concept. This motivates using hierarchical methods to identify latent classes among alternatives in some conceptual representation.

**Hierarchical measures.** An alternate view is that listeners do not reason about alternatives at the level of linguistic forms (i.e., strings), but instead a deeper conceptual level (Buccola et al., 2021). As a proxy for a conceptual representation of an alternative, we use sentence embeddings from Sentence-T5 (Ni et al., 2021). Prior work has shown that clustering over word embeddings has been shown to uncover latent topics, suggesting that there is usable conceptual information represented in the embedding spaces induced by large language models (e.g., Sia et al., 2020; Thompson and Mimno, 2020; Meng et al., 2022). For each sampled alternative $a \in A$, we substitute $a$ into the masked position in the scalar template (Equation (1)) to obtain a full sentence, and then feed this as input to Sentence-T5 to obtain a 768-dimensional embedding of the entire sentence.[4] We assume sentences close in this space are more likely to reflect the same underlying scale, and distant sentences are likely to reflect different scales.

To formalize the idea of conceptual alternatives for scalar inference, we treat scales as latent classes that may give rise to multiple alternative strings. On this view, the surprisal of a strong scalemate is the surprisal of its underlying class, and scale uncer-

---

[2]The completions are not guaranteed to be scalar items, but we take this to be a first approximation. All results are averaged over 4 random seeds for the sampling of alternatives.

[3]https://gist.github.com/sebleier/554280

[4]We use the PyTorch implementation via SentenceTransformers (Reimers and Gurevych, 2019).

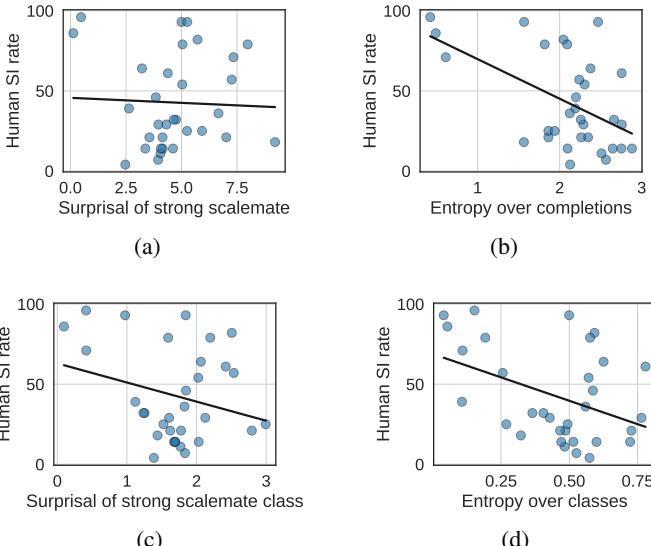

Figure 1: Best-fit linear relationship between human SI rates (van Tiel et al., 2016) and four predictors (Section 3): (a) String-based surprisal of the strong scalemate under T5. (b) Entropy over alternative strings sampled by T5. (c) Surprisal of latent class assigned to the strong scalemate by the Gaussian mixture model. (d) Entropy over probabilities of classes identified by the Gaussian mixture model.

tainty is uncertainty over these classes. To identify latent classes among alternative sentence embeddings, we fit a Gaussian mixture model (GMM) for each set of alternatives (i.e., one per weak scalar item, sentence template, and random seed). To determine the number of latent classes $k$, we fit a GMM for each $k \in \{1, 2, 3\}$ and chose the $k$ that minimized the Bayesian information criterion (BIC) of the fitted model.[5]

After fitting a GMM on the alternative embeddings for each weak scalemate, we predict the class for each alternative. We obtain a score for each class by summing the probabilities assigned by T5 to each alternative within that class. We compute class-based surprisal as the negative log of the score assigned to the class containing the strong scalemate, and class-based scale uncertainty as the entropy over the normalized class scores. As before, we expect that lower surprisal and lower entropy should result in higher SI.

## 4 Results

We computed the four metrics described in Section 3 on the data from Experiment 2 of van Tiel et al. (2016), and evaluated the causal roles of each metric in predicting scalar inference rates across scales. For each of the four metrics, we fit a linear

regression model to predict mean SI rates for each scale (averaged across trials). In all models, we included scale boundedness as an additional predictor, as it is the factor explaining the most variance in van Tiel et al.'s (2016) study.

Our first model tested string-based surprisal as a predictor of SI rates. In line with van Tiel et al.'s results, boundedness is a highly significant predictor ($p < 10^{-16}$). Furthermore, surprisal of the strong scalemate is not a significant predictor ($t = -0.09, p = 0.928$). Figure 1a shows the lack of relationship between in-context surprisal of the strong scalemate and SI rate. Each point represents a scale, with values averaged over the trials and sentence templates (three per scale) presented in van Tiel et al.'s Experiment 2. This lack of relationship concords with van Tiel et al.'s original finding that availability is not predictive of SI rate.

Our second model tested the predictive power of string-based scale uncertainty (i.e., the entropy over completions sampled from T5 in a scalar construction). We found string-based entropy to be a significant predictor of SI rate ($t = -3.28, p = 0.001$), suggesting that uncertainty over alternatives (as string forms) may play a role in scalar inference. Figure 1b shows the negative relationship between SI rates and string-based entropy.

Next, we turn to the hierarchical metrics, which treat alternatives as latent classes in sentence em-

---

[5]For speed of convergence, we assumed diagonal covariance matrices for each estimated class distribution.

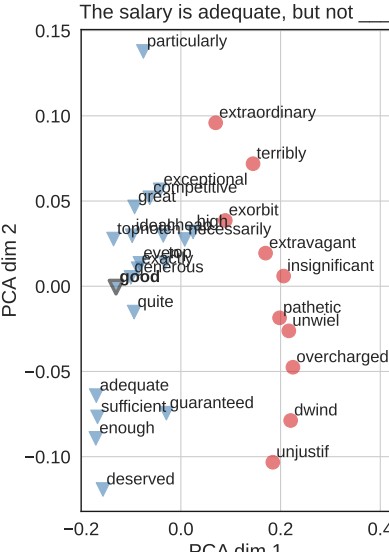

Figure 2: Example of classes (distinguished by color and marker) identified by Gaussian mixture model among alternatives in sentence embedding space. Sentence embeddings are projected into 2 dimensions via PCA for visualization.

bedding space. In general, the pattern mirrors what we found for the string-based metrics. Our third model did not find class-based surprisal to be a significant predictor of SI rates ($t = -1.33, p = 0.186$; Figure 1c), and our fourth model found class-based entropy to be a significant predictor ($t = -2.4, p = 0.01$; Figure 1d).

Finally, we performed a qualitative evaluation of the classes identified by the Gaussian mixture models (GMMs). Figure 2 shows the alternatives generated by T5 for the template "The salary is adequate, but not ____.", with each point obtained by projecting the Sentence-T5 embedding into 2-dimensional space via PCA. The BIC-minimizing GMM identifies two latent classes, distinguished by color and marker, among the alternatives generated by T5 for the weak scalar item *adequate*. First, we examine the cluster containing *good*, the strong scalemate tested in van Tiel et al.'s experiments (marked with boldface and outline). This cluster (indicated by blue triangles) contains *good* as well as semantically similar alternatives such as "great", "sufficient", and "enough". In general, the alternatives in this cluster appear to suggest a scale where high salaries are positive (e.g., from an employee's perspective), with strong scalar items like "generous", "ideal", and "competitive". In contrast, the second cluster (indicated by red circles) contains alternatives such as "extravagant" and "overcharged",

capturing the potential of *adequate* to be on a scale where higher salaries are not always desirable (e.g., from an employer's perspective). While the model-generated alternatives and clusters are noisy, we take this to illustrate that a single weak scalar item (like *adequate*) can plausibly be interpreted as belonging to multiple scales.

## 5 Discussion

We tested the hypothesis that SI rates depend on the listener's confidence in the underlying scale, using two operationalizations of alternatives (surface-level string forms and latent classes in a sentence embedding space). Using data from a previously conducted experiment (van Tiel et al., 2016), we found that scale uncertainty was a significant predictor of SI rates: on average, when uncertainty over alternatives (i.e., entropy over sampled alternatives, or over classes of alternatives in sentence embedding space) is lower, humans are more likely to draw a scalar inference. On the other hand, the predictability of the strong scalemate (as measured by surprisal of the string form, or of its underlying cluster) was not a significant predictor of SI rates.

An open question is why scale uncertainty predicts SI rates, while strong scalemate surprisal and the availability measures from van Tiel et al. (2016) are poor predictors. We conjecture that the predictability of the strong scalemate may be shrouded by the paradigm used in experimental investigations of scalar diversity. In these experiments, the participant is explicitly asked to reason about the strong scalemate in the prompt (e.g., "John says: This student is intelligent. Would you conclude from this that, according to John, she is not brilliant?"). Thus, the effort required to retrieve the strong scalemate (e.g., "brilliant"), which may be captured by its in-context predictability, may no longer be relevant in this setting. We note, however, that our findings likely depend on the chosen clustering algorithm and conceptual representation of the alternatives. We intend to explore this space more broadly in future work.

Looking forward, our methods can be applied to scales that are ordered by ad-hoc relationships instead of entailment (Hirschberg, 1985). Beyond predicting scalar diversity, our approach suggests a way to derive quantitative behavioral predictions from non-linguistic alternatives (Buccola et al., 2021), and supports the idea that context-driven expectations may give rise to pragmatic behaviors.

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
