# OpenReview forum: "Predicting scalar diversity with context-driven uncertainty over alternatives"
_aclweb.org/ACL/2022/Workshop/CMCL — CMCL 2022_

### Official Review · Reviewer_hMpT · 2022-03-22
**Interesting study using language models to explain scalar diversity**

**Rating:** 8
**Confidence:** 4

**Review:**

The paper addresses the linguistic hypothesis that scalar implicatures rates depend on the availability of a strong alternative. They operationalize availability as confidence in the underlying scale and estimate this using neural language modes. The results suggest that such uncertainty over alternatives can explain scalar diversity.

The paper is well-written, with clear explanations of the phenomenon of interest and of the methods used. The methodology introduced is creative and could inspire analogous applications of language models for linguistics.

Other than the correlation values reported, it would be good to have some analysis like a regression, jointly with other potential predictors, in order to clarify the causal role of these uncertainty estimates over the scalar implicature rates.

---

### Official Review · Reviewer_Z16g · 2022-03-24
**Nicely written paper on modeling crucial aspects of scalar implicatures**

**Rating:** 8
**Confidence:** 4

**Review:**

The work reported in these pages tested the hypothesis that scalar implicatures (SI) rates depend on the listener's confidence in the underlying scale, where the availability was estimaed by using T5. The authors reported a significant correlation between SI rates and scale uncertainty.

This paper is well written and sufficently easy-to-read. The methodology is sound and the reported results may follow-up studies.

A few issues:
- the proposed methodology cannot be applied to all kinds of scalar implicatures. For instance, it is difficult to see how this approach can be used to model SIs implying a negation such as the fact that a sentence like "The Sonic Youth played some songs of their fist album" implies that "The Sonic Youth  didn't play all the songs of their fist album". I think that the authors should clearly explain the limitations of their methodology
- by looking at figures 1.b and c it seems that there are a few leverage points that affect the linear relationship between SI rates and entropy or entropy over clusters. How does the correlation score change if we remove these outliers? I'm afraid that it may change significantly, at least in the case of SI ~ Entropy over completions.
- it would be intesteresting to measure the correlation between Entropy over completions and Entropy over clusters.

---

### Official Review · Reviewer_iSz7 · 2022-03-26
**Promising idea, but I had a hard time figuring out exactly what was done**

**Rating:** 6
**Confidence:** 2

**Review:**

This paper describes a model of scalar implicatures that is based on entropy and surprisal, evaluated against human judgements.

It seems like a promising idea, but I had a hard time figuring out exactly what was done.  I suppose the system interprets a weak word to be faint praise if there is a strong word with a high probability in the same context according to T5.  If so, then I suppose the entropy mentioned in the paper is just a high probability of the strong word compared to the weak word?  In any case, I think the presentation would benefit from a simple equation for the estimator and a clear example.

In particular, in the discussion, I did not follow the paragraph describing the example containing the phrase 'if guaranteed has low surprisal when the weak item is possible, then entropy might be low but the surprisal of certain could be high'.  Why does the surprisal depend on the weak item (aren't they alternatives?), and what is the entropy of?

---

### Decision · Program_Chairs · 2022-03-29

Accept